# Semi-Supervised Segmentation of Interstitial Lung Disease Patterns from CT Images via Self-Training with Selective Re-Training

**DOI:** 10.3390/bioengineering10070830

**Published:** 2023-07-12

**Authors:** Guang-Wei Cai, Yun-Bi Liu, Qian-Jin Feng, Rui-Hong Liang, Qing-Si Zeng, Yu Deng, Wei Yang

**Affiliations:** 1School of Biomedical Engineering, Southern Medical University, Guangzhou 510515, China; guangweicai06@gmail.com (G.-W.C.); ybliu1994@gmail.com (Y.-B.L.); fengqj99@smu.edu.cn (Q.-J.F.); 2Department of Medical Imaging Center, Nanfang Hospital of Southern Medical University, Guangzhou 510515, China; lrh1688vip@163.com; 3Department of Radiology, The First Affiliated Hospital of Guangzhou Medical University, Guangzhou 510120, China; zengqingsi@gzhmu.edu.cn

**Keywords:** interstitial lung disease, semi-supervised learning, self-training, segmentation

## Abstract

Accurate segmentation of interstitial lung disease (ILD) patterns from computed tomography (CT) images is an essential prerequisite to treatment and follow-up. However, it is highly time-consuming for radiologists to pixel-by-pixel segment ILD patterns from CT scans with hundreds of slices. Consequently, it is hard to obtain large amounts of well-annotated data, which poses a huge challenge for data-driven deep learning-based methods. To alleviate this problem, we propose an end-to-end semi-supervised learning framework for the segmentation of ILD patterns (ESSegILD) from CT images via self-training with selective re-training. The proposed ESSegILD model is trained using a large CT dataset with slice-wise sparse annotations, i.e., only labeling a few slices in each CT volume with ILD patterns. Specifically, we adopt a popular semi-supervised framework, i.e., Mean-Teacher, that consists of a teacher model and a student model and uses consistency regularization to encourage consistent outputs from the two models under different perturbations. Furthermore, we propose introducing the latest self-training technique with a selective re-training strategy to select reliable pseudo-labels generated by the teacher model, which are used to expand training samples to promote the student model during iterative training. By leveraging consistency regularization and self-training with selective re-training, our proposed ESSegILD can effectively utilize unlabeled data from a partially annotated dataset to progressively improve the segmentation performance. Experiments are conducted on a dataset of 67 pneumonia patients with incomplete annotations containing over 11,000 CT images with eight different lung patterns of ILDs, with the results indicating that our proposed method is superior to the state-of-the-art methods.

## 1. Introduction

Interstitial lung diseases (ILDs) refer to a heterogeneous group of over 200 chronic parenchymal lung disorders that account for 15% of all cases seen by pulmonologists. They are associated with substantial morbidity and mortality. In 2019, there were approximately 655,000 patients affected by ILDs and 22,000 deaths from ILDs in the USA [1]. Considering the wide variety of ILDs, differential diagnosis is necessary to develop appropriate therapy plans for patients with ILDs and to prevent patients from life-threatening complications resulting from misdiagnosis. However, it is fairly difficult even for experienced radiologists due to the inevitable fatigue caused by repetitive work and the similar clinical manifestations between ILDs. Also, it is challenging for existing computer-aided diagnosis (CAD) systems [2] to distinguish the variety of appearances of ILDs in lung CT scans, the most preferred clinical imaging evidence for ILD diagnosis. Developing an automatic and accurate segmentation tool to perform fine-grained classification of ILD patterns from CT images can greatly help radiologists and CAD systems to identify the ILD types, as well as to assess the progression of the diseases by quantitative measures, thereby favoring treatment and follow-up.

Over the past two decades, deep learning methods, especially deep convolutional neural networks (CNNs), have achieved remarkable success in natural and medical image recognition [3,4,5,6]. With the remarkable performance of CNNs in various tasks, they have been gradually adopted for the problem of ILD pattern recognition [7,8,9,10,11,12,13,14]. Most previous work often focused on image-level ILD identification [11,12] and patch-level ILD pattern classification [7,8,9,10,14]. The image-level ILD identification methods assign single or multiple labels to a holistic CT slice with ILDs, but they cannot locate ILD lesions in CT images for quantitative analysis. Patch-level ILD pattern classification methods often classify small regular image patches or regions of interest (ROIs) (i.e., 32×32) into a specific ILD pattern. However, the image patch size is relatively small, where some visual details and spatial context may not be fully captured. The pre-defined regular patches or ROIs for detection remain not fine-grained enough, which tends to result in a misclassification of many different classes of pixels near the boundary. This is because the spread of the disease is not symmetric but arbitrary and multiple ILD patterns may coexist on a CT patch. Therefore, pixel-level segmentation of ILD patterns from CT images is more desirable, but is challenging for some of the reasons given below. First, ILD patterns are diverse and hard to distinguish in CT images due to inter- and intra-class variability. Figure 1 shows eight typical ILD patterns with totally different textures. Second, obtaining a large amount of well-labeled data is extremely expensive due to the expert-driven and time-consuming nature of pixel-level ILD annotations from CT scans. As far as we know, there are only a few studies that pixel-by-pixel segment the ILD patterns in CT images [13]. Anthimopoulos et al. in [13] proposed a dilated fully convolutional network for segmenting six typical ILD patterns on the publicly available multimedia database of interstitial lung diseases maintained by the Geneva University Hospital (HUG) [15]. However, this database was sparsely annotated, i.e., only labeling one prominent disease region, not an entire CT slice. This imposes limitations on the segmentation models that need to discard unlabeled slices during training and exclude unannotated lung regions during evaluation.

To leverage the unlabeled CT slices in our collected dataset and boost the segmentation performance, we propose the adoption of a popular semi-supervised learning (SSL) framework termed Mean-Teacher in our work, which was initially proposed for image classification [16]. A typical Mean-Teacher framework consists of a teacher model and a student model. The framework aims to improve the student model’s predictions by using the teacher model’s predictions as a guide. This is accomplished by applying a consistency loss between the predictions of the two models to encourage consistent outputs. To better capture the discriminative features of ILD patterns in CT images, we select a high-resolution network (HRNet) with parallel multi-resolution representations as the network of the two models. In SSL, self-training approaches have been widely applied to improve segmentation performance. For example, the student model is retrained using the pseudo-labeled data generated by the teacher model from unlabeled data. However, since the teacher model cannot predict well on all unlabeled data, it may result in potential performance degradation when iteratively optimizing the model with bad pseudo-labels. To mitigate this issue, a selection procedure is necessary to select reliable pseudo-labels for retraining the student model. In our work, we adopt the latest selection strategy as in ST++ [17] to select and prioritize more reliable and easier images in the re-training phase.

To sum up, we propose an end-to-end semi-supervised learning framework for the segmentation of ILD patterns (ESSegILD) from CT images via self-training with selective re-training. The main contributions of our work can be summarized as follows:(1)We propose a novel method termed ESSegILD for the segmentation of ILD patterns from CT images. As far as we know, this is one of the few studies on pixel-level ILD pattern recognition in CT images.(2)In our proposed ESSegILD framework, we utilize consistency regularization and self-training with selective re-training to appropriately and effectively utilize the unlabeled images for improving the segmentation performance. Therein, a high-resolution network (HRNet) with parallel multi-resolution representations is adopted as the backbone of our model to better capture the discriminative features of ILD patterns.(3)Extensive experiments are conducted on a large-scale partially annotated CT dataset with eight different ILD patterns, with results suggesting the effectiveness of our proposed method and its superiority to other comparison methods. To the best of our knowledge, the ILD patterns identified in our work are the most diverse ever reported.

## 2. Related Work

### 2.1. ILD Pattern Recognition

ILDs are characterized by textural changes in the lung parenchyma, which are often assessed using texture classification schemes on local regions or volumes of interest [18,19]. In recent years, solutions based on CNNs have been proposed for lung pattern recognition in ILDs. However, the majority focused on the patch-level or image-level classification of ILDs from CT images [10,11,20,21,22,23,24]. Patch- or image-level recognition usually classifies the regular image patches or the whole slices into a class of the ILD patterns from manually annotated polygon-like regions of interest (ROIs) or 2D axial slices. The classification often tends to misclassify many different classes of pixels in a single CT slice. Although recent approaches have utilized deep learning techniques for the automatic detection or identification of ILDs [25,26,27,28], the development of pixel-level segmentation models is hindered by the limited availability of labeled data and high labor costs. Consequently, there are only a few studies focused on pixel-level recognition for ILD patterns from CT images. The most related work to ours is [13], which introduced a deep dilated CNN for the semantic segmentation of ILD patterns. However, the training data used in their work were sparsely annotated, i.e., not covering the entire CT slices or lesion regions but the most typical disease regions. In this paper, we collect a relatively large dataset of 67 pneumonia patients comprising more than 11,000 CT images with a slice-wise sparse annotation, i.e., only labeling a few slices in each CT volume with ILD patterns pixel by pixel to minimize the annotation burden on radiologists. Then, we present a semi-supervised segmentation method for the pixel-by-pixel recognition of multiple ILD patterns from CT images using the collected CT scans with partial annotations.

### 2.2. Semi-Supervised Learning

Semi-supervised learning (SSL) has demonstrated promising results in the field of image classification/segmentation by utilizing unlabeled data to promote model training. Two common strategies employed in previous SSL approaches are consistency regularization [16,29,30,31,32,33,34,35,36,37,38,39] and self-training [17,40,41,42,43,44,45,46].

Consistency regularization involves enforcing the consistency of predictions with various perturbations, i.e., performing data or model perturbations while enforcing consistency among the predictions. This idea was used in [29], where a temporal ensembling method was proposed to form a consensus prediction of the unknown labels using the outputs of the network in training on different epochs under different regularization and input augmentation conditions. Later, Mean-Teacher was proposed, averaging model weights instead of label predictions [16], showing better performance than the temporal ensembling method. In a Mean-Teacher framework consisting of a teacher and a student network, the teacher network has a similar architecture to the student network, but its parameters are updated as an exponential moving average (EMA) of the student network weights. The output of the student is compared with that of the teacher using consistency loss. Inspired by the impressive performance achieved by Mean-Teacher, many studies have extended it to improve model performance [30,31]. For instance, Miyato et al. [30] presented a novel data augmentation method via virtual adversarial learning (VAT) and applied it to the input, where consistency regularization was imposed on the predictions. Yu et al. [33] applied the Mean-Teacher paradigm to the task of semi-supervised 3D left atrium segmentation and introduced an uncertainty estimation method to guide the calculation of consistency loss.

Self-training seeks to generate pseudo-labels for unlabeled data and use them to expand the labeled training data to train more powerful models. This idea was used in [47], where they selected the classes with the maximum predicted probability and used them as if they were true labels to train a semi-supervised classification model. Different strategies have been developed to reduce the adverse effect of unreliable pseudo-labels caused by limited labeled data [40,41,43]. More recently, Yang et al. [17] proposed an advanced self-training framework (namely, ST++) to select reliable pseudo-labeled images based on holistic prediction-level stability. The improved performance indicates the efficacy of their idea that exploits the pseudo-labeled images in a reliable-to-unreliable and easy-to-hard curriculum manner.

Inspired by previous work, our proposed ESSegILD method adopts the Mean-Teacher model as a foundation, together with the latest ST++ technique [17], for the semi-supervised segmentation of ILD patterns from CT images. We aim to make maximum use of the unlabeled data from two aspects, i.e., consistency regularization and self-training.

## 3. Dataset and Methods

In this work, we have collected a dataset of 67 CT scans, which comprises over 11,000 CT images of eight different kinds of ILD patterns. In addition, we have obtained pixel-level annotations for a small fraction of the images, which provides us with a valuable set of labeled examples to train and evaluate our algorithm. Using the ILD dataset with partial annotations, we developed an end-to-end semi-supervised learning method for segmenting ILD patterns from CT images. Specifically, ESSegILD, which aims to leverage the partially labeled dataset from CT images via self-training with selective re-training, is proposed to reduce the dependence on expensive and time-consuming manual annotations. In the following, we will describe the dataset and our proposed method.

### 3.1. Dataset

Clinical high-resolution CT scans from 67 pneumonia patients were collected from the First Affiliated Hospital of Guangzhou Medical University from 2015 to present. All cases were clinically diagnosed as pulmonary alveolar proteinosis (PAP), usual interstitial pneumonia (UIP), or lymphangiomatosis (LAM) with ILD lesions. The number of slices in each CT scan ranged from 127 to 339, with a resolution of 512×512 and a pixel size ranging from 0.52 to 0.67 mm. To ensure the quality and relevance of our dataset, we selected cases that met three inclusion criteria: (a) the subjects had to be diagnosed with ILDs in multiple disciplines; (b) the CT scans were acquired by spiral CT and reconstructed by a high-resolution algorithm, with slice thicknesses of 1–2 mm; (c) there were no other concurrent lung diseases, such as infection, pneumoconiosis, tumor, etc. Then, to generate ground-truth segmentation masks, two radiologists with 7 and 19 years of experience were invited to annotate the ILD patterns in the dataset. Notably, a slice-wise with pixel-level sparse annotation method was adopted in our work, i.e., only labeling a few slices for each CT scan and the annotation covers the entire lung region for the slices selected by the two radiologists. This kind of annotation could alleviate the labeling burden on radiologists as well as increase the diversity of the labeled dataset. In CT images, the features of adjacent slices are often highly correlated, and lesions typically appear consecutively across adjacent slices, so we selected unlabeled slices from 3 upper and lower areas of the labeled slice as unlabeled data to train our model. In such a way, a total of 11,493 CT images are included in our work, of which 2132 images are fully annotated. All comparison methods follow the same data partition divided on the patient basis, including 1732 labeled and 7360 corresponding unlabeled images as the training set, 200 labeled images as the validation set, and 200 labeled images as the test set. All annotated areas contain 8 types of lung patterns with ILDs, including Healthy (H), Honeycombing (HC), Pneumothorax (Pne), Cyst, Linear (Line), Ground-glass opacity (GGO), Reticular opacity (RO), and Consolidation (Cons).

### 3.2. Methods

In this work, we propose an **e**nd-to-end **s**emi-supervised learning method for **seg**menting **ILD** patterns from CT images (denoted as ESSegILD). As shown in Figure 2, we propose the adoption of a popular semi-supervised model, i.e., *Mean-Teacher*, as our model, which includes a teacher model and a student model. Inspired by [17], a selective re-training strategy is introduced to leverage a wealth of information from unlabeled images during training our ESSegILD model. The selective retraining strategy allows the reliability of those pseudo-labels generated by the teacher model to be ranked and more reliable pseudo-labeled data to be prioritized to expand the labeled dataset for re-training the model. Such a training process is repeated to progressively improve the model performance until it remains stable.

#### 3.2.1. The Proposed ESSegILD Framework

For densely segmenting the ILD patterns from CT images in a semi-supervised setting, we collected a relatively large CT dataset, with slice-wise sparse annotations for each CT volume, for building a combination set of manually labeled images $D^{l}={\left(x_{i},y_{i}\right)}_{i=1}^{M}$ and unlabeled images $D^{u}={\left(u_{i}\right)}_{i=1}^{N}$, where N>M. Via pseudo-labeling, high-confidence pseudo-labeled images $D^{p}={\left(u_{i},y_{i}^{p}\right)}_{i=1}^{K}$ are selected to expand the labeled set $D^{l}={\left(x_{i},y_{i}\right)}_{i=1}^{M+K}$. As presented in Figure 2, an iterative re-training strategy is utilized in the proposed method, in which the model is retrained in a new training round using the updated dataset consisting of expanded labeled data and reduced unlabeled data. The training process of our proposed ESSegILD method includes four steps as described below:Step 1:Training the segmentation model *f* using labeled set $D^{l}$ and unlabeled set $D^{u}$.Step 2:Pseudo labeling $D^{u}$ using the pretrained teacher model and selecting high-confidence pseudo-labeled data from $D^{u}=\left(u_{i},f(u_{i})\right)$ to obtain pseudo-labeled set $D^{p}=\left(u_{i},p_{i}\right)$, where the model’s prediction $f(u_{i})$ is represented as $p_{i}$.Step 3:Updating the labeled and unlabeled images in the training set by $D^{l}$ = $D^{l}+D^{p}$ and $D^{u}$ = $D^{u}-D^{p}$.Step 4:Retraining the model *f* using the updated dataset.Step 5:Repeating Steps 2–4 until reaching the maximum training rounds.

Here, we describe how to train the segmentation model and select high-confidence pseudo-labeled samples in the training process.

*Model training.* In the proposed ESSegILD method, we adopt the popular semi-supervised model, i.e., Mean-Teacher, as our segmentation model. Thus, our segmentation model consists of a teacher model and a student model as in typical Mean-Teacher models. Note that only the student model is involved in the backward propagation to update its weights, while the weights of the teacher model are updated as an exponential moving average (EMA) of the student weights according to the following equation:(1)$$\theta_{k}^{*}=\alpha {\theta^{*}}_{z-1}+\left(1-\alpha \right)\theta_{z}\;$$
where $\theta^{*}$ and θ are the parameters of the teacher and student models, respectively, *z* is the training step, and α∈(0,1) is a smoothing coefficient. For training the student model, a supervised loss term for labeled data and a consistency loss term for labeled and unlabeled data are used to optimize the training process. As we know, ILD lesions tend to be diffuse and irregular in shape, and different ILD patterns can occupy distinct regions in lung CT slices, resulting in the imbalance of the training data. A combination of dice loss [48] and focal loss [49] is used as the supervised loss term $L_{sup}$ for labeled data to mitigate the imbalance issue, which can be expressed as
(2)$$L_{sup}=\left(1-\rho \right)L_{Dice}+\rho L_{FL}$$
where ρ is used to balance the relative importance of dice loss and focal loss. Therein, $L_{Dice}$ and $L_{FL}$ can be formulated as
(3)$$L_{Dice}=-\sum_{t}^{T}\frac{{\left(p_{t}-y_{t}\right)}^{2}}{p_{t}^{2}+y_{t}^{2}}$$
(4)$$L_{FL}=-\sum_{t}^{T}\beta_{t}{\left(1-h_{t}\right)}^{\gamma }\log\left(h_{t}\right)$$
(5)$$h_{t}=\left\{\begin{array}{c} p_{t}\;\;\;\;\;\;\;\;\;\;\;if\;y_{t}=1\\ 1-p_{t}\;\;\;\;\;otherwise \end{array}\right.$$

In Equation (Equation 3), $p_{t}$ denotes the probability that a pixel belongs to class *t*, $y_{t}$ represents the corresponding ground-truth binary label of class *t*, and *T* is the number of categories to be identified. In Equation (Equation 4), $\beta_{t}$ is used to balance the importance of different samples across different categories, which depends on the amount of data available for each category. The variable $h_{t}$, as defined in Equation (Equation 5), represents the probability of the model correctly predicting the sample, and ${\left(1-h_{t}\right)}^{\gamma }$ is the modulating factor to encourage the model to prioritize challenging samples over easier ones by assigning a higher weight to the challenging samples. The modulating factor can be modified to control the contribution of easy samples to the loss function by adjusting the value of γ.
(6)$$L_{con}=\left|\right|f\left(x,\theta^{*}\right)-f\left(x,\theta \right){\left|\right|}^{2}\;$$

A consistency loss is often used to force the teacher and the student to make stable and consistent predictions on labeled and unlabeled data under various perturbations. We adopt the mean square error (MSE) in Equation (Equation 6) as the consistency loss, where $f(x,\theta^{*})$ and f(x,θ) correspond to the predictions of the teacher and student models, respectively. Thus, the total loss function of the segmentation model $L_{total}$ can be expressed as
(7)$$L_{total}=L_{sup}+\lambda *L_{con}$$
where λ is the hyper-parameter to balance the two loss terms, i.e., $L_{sup}$ and $L_{con}$.

*High-confidence pseudo-labeled sample selection.* To reduce the bias by unreliable pseudo-labels, an effective selection step is necessary to select high-confidence pseudo-labeled samples from an unlabeled set. Yang et al. [17] found that there was a positive correlation between the segmentation performance and the evolving stability of produced pseudo masks during the training phase. Inspired by ST++ [17], we introduce filtering out unreliable pseudo-labeled images from an unlabeled set $D^{u}$ at an image level based on their evolving stability during training. Specifically, considering an unlabeled image $u_{i}\in D^{u}$, we predict the pseudo masks of $u_{i}$ with *Q* checkpoints ${\left\{f_{j}\right\}}_{j=1}^{Q}$ of the teacher model saved during training to obtain $\left\{Y_{ij}^{p}\right\}$. Then, a stability score for an unlabeled image $u_{i}$ can be calculated by
(8)$$\begin{array}{cc} s_{i} & =\sum_{j=1}^{Q-1}meanIOU\left(Y_{ij}^{p},Y_{iQ}^{p}\right)\\ & =\sum_{j=1}^{Q-1}\frac{Y_{ij}^{p}\cap Y_{iQ}^{p}}{Y_{ij}^{p}\cup Y_{iQ}^{p}} \end{array}$$

In this work, we sort the scores in descending order and prioritize unlabeled samples with more reliable pseudo-labels that have higher stability scores and select them to expand the labeled set $D_{l}$ to retrain the segmentation model in the next training round.
(9)$$Dice=\frac{2|X\cap Y|}{\left|X|+|Y\right|}$$

For quantitative evaluation, we used the dice coefficient in Equation (Equation 9) as the principal performance metric. In this equation, *X* and *Y* represent the region segmented by the trained teacher model and the ground truth, respectively.

#### 3.2.2. High-Resolution Network

The majority of existing medical image segmentation networks, such as U-Net [50] and SegNet [51], consist of a contracting path and an expanding path, with the former encoding the input image as low-resolution feature maps to capture semantic information and the latter recovering the feature maps to the original resolution for precise localization. However, it will lose information, especially for fine structures, due to the unidirectional down- and up-sampling processes. As the imaging features of ILD patterns in HRCT images are very similar, the detailed information of the feature map is essential to distinguish the similar manifestations of ILD patterns. To better capture the discriminative features of ILD patterns, a high-resolution network (HRNet) with parallel multi-resolution representations [52] is adopted as the basic network for both the teacher and student networks in our ESSegILD model. The HRNet network has the following features: (1) connecting the high-to-low-resolution convolution streams in parallel, and (2) repeatedly exchanging information across resolutions. In such a way, the resulting representation is semantically richer and spatially more precise by maintaining high-resolution representation throughout the entire training process. More details about the network architecture of HRNet can be found in [52].

### 3.3. Implementation Details

*Image pre-processing.* Lung segmentation and intensity normalization were first performed on the CT images. We carefully followed the previous image pre-processing method in [10], and the image intensity within the extracted lung region was cropped to a specific Hounsfield unit (HU) window of [−1000, 250] and mapped to [0, 1]. Before feeding the input CT images into the segmentation model, we first augmented them by adding noise, flipping, rotation, and shifting.

*Network training.* To reduce the effect of the student on the teacher when the student’s performance is poor, we set the value of α=0.99 in Equation (Equation 1) to obtain a more robust teacher network during the ramp-up phase and α=0.999 for the rest of the training. To minimize the confirmation bias and enhance the quality of pseudo-labels, we used a batch size of 4, consisting of 1 labeled slice and 3 unlabeled slices, and added different noise perturbations to the input data during each forward propagation process in both the teacher and student networks. We also took a dropout ratio of 0.5 at the end of the teacher and student networks as a method of model perturbation for better performance. We set the value of γ=2 in Equation (Equation 4) to alleviate the effect of unbalanced data and ρ=0.75 in Equation (Equation 2) to make the model focus on hard-to-classify samples. At the beginning of the training, we expected the network to focus on labeled data to accelerate the convergence of the network, so the λ in Equation (Equation 7) increased gradually from 0 to 1 as the training steps increased during each round of model re-training. At the end of every training round, the stability score for each unlabeled image $s_{i}$ in Equation (Equation 8) was computed. We set Q=4 and simply treated the top 25% highest-scored images as the reliable ones to expand the labeled set to train the model in the next training round. In one training round, the total training epochs of the segmentation model were 200 and the training took about 8 h on a machine equipped with a GPU NVIDIA GeForce RTX 2080 Ti.

## 4. Experiments and Results

### 4.1. Experimental Setup

In this section, we conduct an ablation study to validate the efficacy of different key factors in our proposed ESSegILD, and a group of experiments to compare our method with several recent semi-supervised segmentation methods, including MT [16], UA-MT [33], CPS-Seg [53], ST [47], and ST++ [17]. Here, we briefly introduce these comparison methods.

*MT.* Here we refer to the classical *Mean-Teacher* method as MT for short. The MT method proposes a consistent regularization loss for both labeled and unlabeled data to constrain the teacher and student models to output similar results for the same input under different perturbations. Different from our ESSegILD method, the MT model needs to be trained only once without iterative re-training. *UA-MT.* The UA-MT method has been proposed for semi-supervised 3D left atrium segmentation by Yu et al. [33]. It also adopts the Mean-Teacher framework, and an uncertainty-aware consistent loss is presented in the method to mitigate the adverse effects of unreliable predictions by the teacher model.

*CPS-Seg.* The CPS-Seg method has been originally proposed for semi-supervised natural image segmentation, in which a novel consistency regularization approach, called cross-pseudo supervision (CPS), is introduced to impose consistency on two segmentation networks.

*ST.* The self-training (ST) method via pseudo-labeling is a simple and efficient semi-supervised learning method. The proposed network is trained in a supervised manner using labeled data and pseudo-labels simultaneously.

*ST++.* Most recently, Yang et al. [17] proposed an advanced ST++ framework based on ST, in which more reliable images are automatically selected and prioritized in the re-training phase. Inspired by ST++, we also progressively leveraged unlabeled data via selective re-training in our method.

### 4.2. Ablation Study

Three key factors, including network architecture, training data, and iterative re-training, mainly affect the segmentation performance in our proposed ESSegILD method. To investigate the impact of different factors, we have conducted an ablation study with alternative configurations.

*Impact of Network Architectures.* We trained different segmentation models to identify different ILD patterns in CT images using different networks, such as U-Net [50], Dilated CNN [13] and HRNet, and compared their segmentation performance to validate the efficacy of the HRNet as the basic network in our method. The Dilated CNN network is the same as in [13], in which it is used for the semantic segmentation of ILD patterns. Notably, we only used the labeled data from our collected dataset to train these models in a supervised manner for rapid validation. The segmentation results for 8 ILD patterns of different segmentation models using various networks are reported in Table 1. In Table 1, we can see that the segmentation performance of the HRNet network is superior to other models using the network architecture of U-Net and Dilated CNN. The competitive segmentation results demonstrate the superiority of high-resolution representation used in the HRNet for capturing the discriminative features of ILDs in CT images. Therefore, we adopted HRNet as the basic network of our teacher and student models in our ESSegILD framework.

*Impact of Using Different Training Data.* As we mentioned before, we collected a dataset containing 11,493 CT images from 67 subjects with eight ILD patterns, among which only a small fraction of CT images were fully annotated. For training our ESSegILD model, we leveraged the information from both the labeled data and the remaining mostly unlabeled data as the training data to boost the segmentation performance in a semi-supervised learning manner. To explore the gain in segmentation performance of the unlabeled data in our training data, we trained our model using the labeled data and the remaining unlabeled data in different proportions, such as 25%, 50%, 75%, and 100%. When only using those 1732 fully annotated images, a supervised-only (SupOnly) model was trained as the baseline model, in which the supervised loss functions, such as dice loss and focal loss, and the network architecture, i.e., HRNet, were consistent with our proposed ESSegILD model. Table 2 reports the comparison of the SupOnly model and our semi-supervised ESSegILD models with the different amounts of unlabeled training data. Note that the number of training rounds for all semi-supervised ESSegILD models in Table 2 was set to three. From Table 2, we can make two observations. *First*, as more unlabeled data are included in the training data, the segmentation performance of the corresponding model is improved. *Second*, our ESSegILD model with labeled and unlabeled data largely improves the segmentation performance of the SupOnly model by 2.36 percent points regarding the average dice coefficient of eight ILD patterns. These results suggest that our proposed semi-supervised learning framework can boost the segmentation performance by effectively leveraging unlabeled data.

*Impact of Iterative Re-training.* In our work, we adopted the latest selective re-training strategy [17] to select reliable pseudo-labeled samples as if they were true labeled data to re-train our ESSegILD model. To further boost the performance, we assigned pseudo labels to unlabeled images using the current teacher model, selected reliable pseudo-labeled data to expand the labeled dataset, and then re-trained the model using the updated dataset, which was repeated several times. Intuitively, the model performance improves as the number of training rounds increases, but it also increases the training time. Therefore, a proper number of training rounds is required to balance the training time and segmentation performance. Figure 3 shows the relationship between the segmentation metric (i.e., the average dice of segmenting eight lung patterns) and the number of training rounds. From Figure 3, we can observe that the segmentation performance improves as the number of training rounds increases. Retraining the model more than three times makes it slower to improve the model’s performance. To balance efficacy and efficiency, the number of training rounds was set to three in this work.

### 4.3. Comparison with the State-of-the-Art Semi-Supervised Segmentation Methods

We implemented several state-of-the-art semi-supervised segmentation methods for comparison. Note that we used the same network (HRNet) in these methods for a fair comparison. Table 3 shows the quantitative results of different comparison methods, including MT, UA-MT, CPS-Seg, ST, ST++, and our proposed ESSegILD, for segmenting eight ILD patterns from high-resolution CT images. As shown in Table 3, in the comparison between two self-training methods, i.e., ST and ST++, the ST++ is better than ST since it selects the trustworthy pseudo-labels as if they were true labels to augment the labeled dataset and then trains the model in a progressive way. This shows the effectiveness of the selective re-training strategy in [17] for improving the performance of semi-supervised learning. Inspired by the improvement achieved by ST++, our proposed ESSegILD combines the selective re-training strategy and consistency regularization to better leverage abundant unlabeled data. As shown in Table 3, our proposed ESSegILD achieves the best performance regarding the dice coefficient of all lung patterns. Specifically, the ESSegILD surpasses methods based on improved consistency regularization, such as UA-MT and CPS-Seg, as well as the self-training methods (i.e., ST and ST++) without consistency regularization, which suggests the efficacy of the combination of consistency regularization and selective re-training in semi-supervised learning. We also show the local and global visual results of all comparison methods in Figure 4 and Figure 5. Consistent results can be seen in them. The predicted masks from our proposed ESSegILD are the closest to the manually outlined masks. Several strategies may contribute to the better segmentation results of our method. Previous semi-supervised segmentation approaches typically focus on training models using large-scale natural images, which may not perform optimally on CT images with ILDs. Additionally, factors such as the selection of appropriate loss functions, criteria for selecting pseudo-labels, and the selection ratio of pseudo-labeled data for retraining can also significantly impact the final segmentation results. In the proposed ESSegILD, a better pretrained teacher model can generate higher-quality pseudo-labels at the beginning of the iterative re-training process, which may also be a key factor in the better experimental results.

## 5. Discussion

For dense medical image segmentation, costly pixel-wise labeling and required expert domain knowledge make it extremely difficult to collect large amounts of labeled data, thus hindering the development of data-driven deep learning models for automatic segmentation. Semi-supervised learning (SSL) can mitigate the requirement for labeled data by incorporating unlabeled data into model training. In this work, we collected a dataset of 67 pneumonia patients containing over 11,000 CT images with eight typical ILD patterns and propose an effective semi-supervised framework that integrates consistency regularization and self-training (or pseudo-labeling), called ESSegILD, for recognizing multiple ILD disease patterns in CT images. The proposed model can automatically segment ILD lesions in CT images, consequently enhancing the diagnostic performance of radiologists. We believe that it can significantly alleviate the burden on physicians and contribute to patients’ treatment and follow-up.

In our work, we performed an extensive ablation study and a group of method comparison experiments to validate the efficacy of the proposed ESSegILD. Several strategies contributed to the better segmentation results of our method. First, the basic network that is adopted in our work, i.e., HRNet with parallel multi-resolution representations, is powerful enough to effectively learn knowledge from entire CT input images. Second, we propose an effective way to utilize unlabeled data at scale. Some other strategies, such as the appropriate loss functions and the election ratio of pseudo-labels in the selective re-training data, may also be potential reasons to boost the segmentation performance. However, our main point lies in progressively leveraging unlabeled images by selecting and prioritizing more reliable and easier images in the re-training phase and encouraging the consistency of the predictions of different networks for the same input.

Although our proposed method can take advantage of unlabeled images, it does not take into account the inter-slice consistency of a CT volume, i.e., the features of adjacent slice images in a CT volume are highly correlated and lesions usually appear consecutively between adjacent slices. Moreover, the selection of pseudo-labels in our method may inevitably include some poor-quality unlabeled samples, limiting the model’s potential to achieve better results. To address these limitations, inter-slice consistency information and potential poor-quality pseudo-labeled samples should be effectively utilized for segmenting multiple ILD patterns. In the future, we will explore the method to utilize the inter-slice consistency information and make full use of the potential poor-quality pseudo-labels. We aim to integrate the model into a CAD system to assist radiologists in the differential diagnosis of ILDs.

## 6. Conclusions

In this paper, we develop an end-to-end semi-supervised segmentation method for ILD pattern recognition, called ESSegILD. The results of extensive ablation studies and comparison experiments have demonstrated the effectiveness of our approach. Specifically, the adopted HRNet, which maintains high-resolution representation throughout the entire training process, enhances the ability of the network to capture the discriminative features of ILD patterns. The iterative re-training strategy can select more reliable pseudo-labels to boost segmentation performance, demonstrating its effectiveness in progressively expanding the labeled set for subsequent training rounds. Finally, the experimental results of the proposed ESSegILD with consistency regularization and self-training show significant performance improvements with respect to the state-of-the-art semi-supervised segmentation methods. In our future research, we will focus on addressing the limitations of the current method to boost segmentation performance by utilizing inter-slice consistency information and leveraging poor-quality pseudo-labeled samples.

## Figures and Tables

**Figure 1 bioengineering-10-00830-f001:**
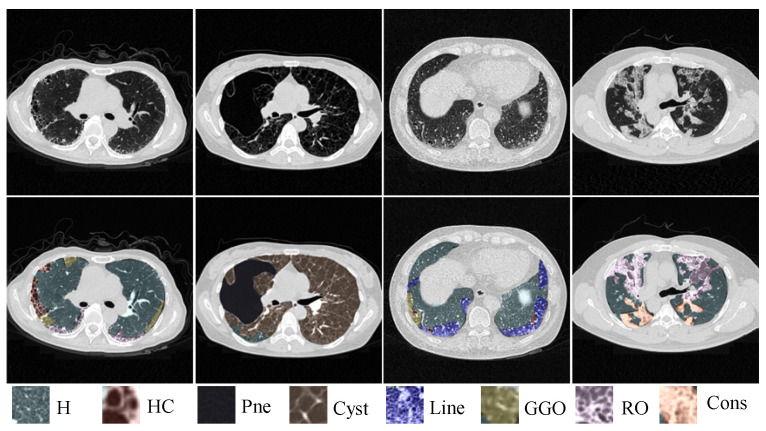
Illustration of similar manifestations with different ILD patterns coexisting on a single CT slice. ‘H’, ‘HC’, ‘Pne’,‘Cyst’, ‘Line’, ‘GGO’, ‘RO’ and ‘Cons’ represent Healthy, Honeycombing, Pneumothorax, Cyst, Linear, Ground-glass opacity, Reticular opacity and Consolidation, respectively.

**Figure 2 bioengineering-10-00830-f002:**
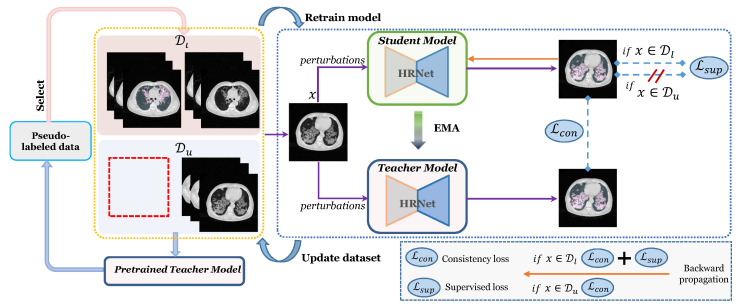
Illustration of our proposed ESSegILD framework for segmentation of ILD patterns from CT images. The framework consists of three components: the training of Mean-Teacher model with labeled images $D_{l}$ and unlabeled images $D_{u}$, pseudo-label selection, where a stability score is calculated for the selection of reliable pseudo-labels, and iterative re-training for the retraining of the Mean-Teacher using the updated dataset.

**Figure 3 bioengineering-10-00830-f003:**
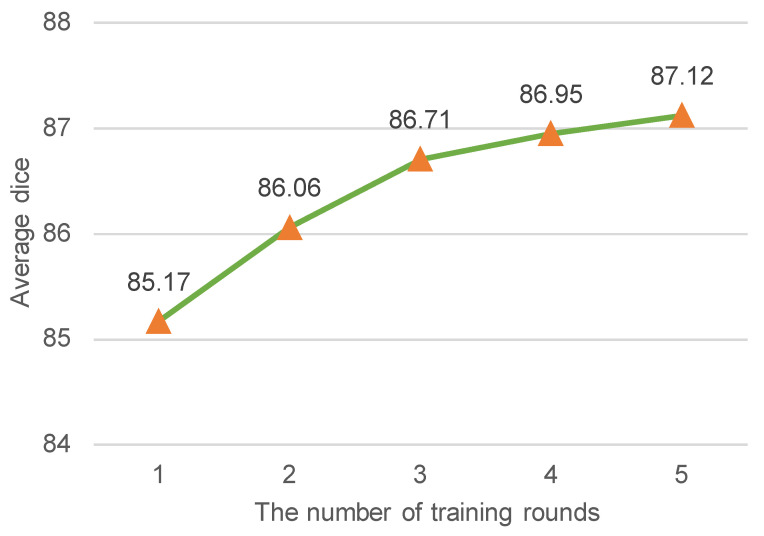
The average dice coefficient (%) vs. the number of training rounds. The average dice indicates the mean segmentation performance of the model for 8 ILD patterns. Intuitively, the improvement of dice coefficient becomes less pronounced after a certain number of training rounds. To achieve a balance between segmentation performance and training time, the training rounds were limited to 3 in this study.

**Figure 4 bioengineering-10-00830-f004:**
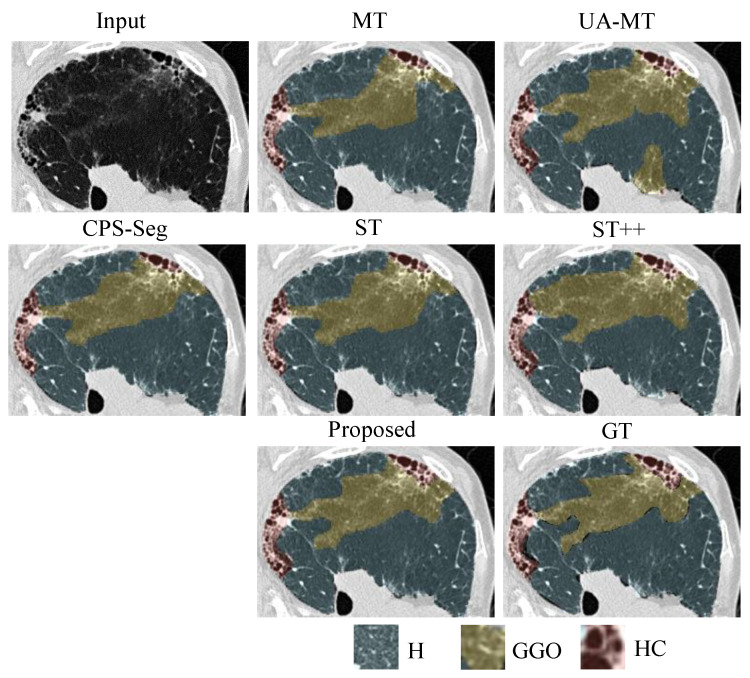
Local visual results of different methods for segmenting ILD patterns from CT images. Three types of ILD patterns (Healthy/Ground-glass opacity/Honeycombing) coexist in the partial CT slice. We can see that the predicted masks from our proposed ESSegILD are the closest to the ground-truth labels (GT).

**Figure 5 bioengineering-10-00830-f005:**
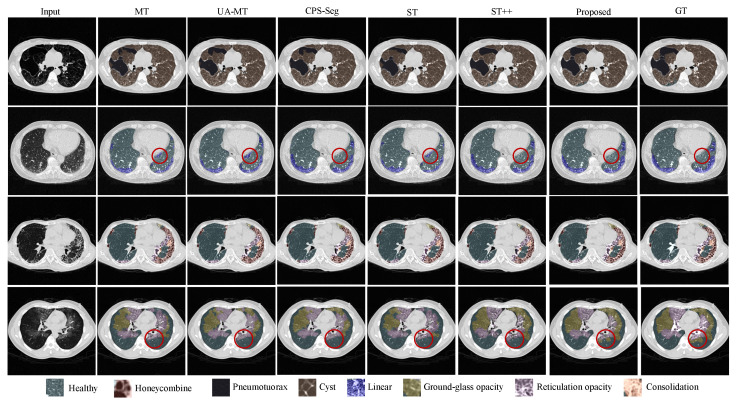
Global visual results of different methods for segmenting 8 ILD patterns from CT images. For better comparison, some regions in the segmentation result of each method are marked with red circles. The proposed method demonstrates improved responses in these regions compared to the ground truth.

**Table 1 bioengineering-10-00830-t001:** Segmentation performance of different network architectures, measured by dice coefficient (mean ± s.d.%). The dice coefficient for each pattern is calculated separately using Equation (Equation 9). The term Avg in the table denotes the average dice score of all patterns. The networks are trained with only labeled data in a supervised manner. The bold font represents the optimal experimental result.

Network	H	Con	HC	Pne	RO	Line	GGO	Cyst	Avg
U-Net	86.26 ± 2.24	83.02. ± 3.08	82.76 ± 3.09	93.08 ± 1.43	77.17 ± 3.45	84.44 ± 2.62	72.29 ± 4.17	94.77 ± 1.17	82.68 ± 6.52
Dilated CNN	86.73 ± 2.03	83.62 ± 2.91	83.08 ± 2.85	93.49 ± 1.29	78.59 ± 3.36	85.93 ± 2.45	73.45 ± 3.97	95.13 ± 1.06	83.50 ± 6.34
HRNet	**87.81 ± 1.90**	**84.83 ± 2.82**	**84.39 ± 2.63**	**94.12 ± 1.17**	**79.06 ± 3.28**	**86.97 ± 2.08**	**73.93 ± 3.83**	**95.40 ± 0.94**	**84.35 ± 6.19**

**Table 2 bioengineering-10-00830-t002:** Comparison of our segmentation models when additional different proportions of unlabeled data are included during training. The SupOnly model is trained in a supervised manner using only labeled data, while the other variants are trained in a semi-supervised manner using both labeled and different proportions of unlabeled data. The bold font represents the optimal experimental results achieved by our model with different proportions of unlabeled data. All results are measured by dice coefficient (mean ± s.d.%).

Training Data	H	Con	HC	Pne	RO	Line	GGO	Cyst	Avg
Labeled (SupOnly)	87.81 ± 1.90	84.83 ± 2.82	84.39 ± 2.63	94.12 ± 1.17	79.06 ± 3.28	86.97 ± 2.08	73.93 ± 3.83	95.40 ± 0.94	84.35 ± 6.19
Labeled + 25% Unlabeled	88.29 ± 1.69	85.31 ± 2.54	84.73 ± 2.56	94.62 ± 1.12	79.64 ± 3.17	87.61 ± 1.94	74.78 ± 3.71	95.49 ± 0.93	84.79 ± 6.08
Labeled + 50% Unlabeled	89.51 ± 1.45	86.08 ± 2.23	85.64 ± 2.39	95.37 ± 1.06	80.78 ± 2.92	88.51 ± 1.73	76.25 ± 3.50	95.88 ± 0.88	85.98 ± 5.67
Labeled + 75% Unlabeled	89.93 ± 1.38	86.42 ± 2.15	86.27 ± 2.24	95.58 ± 0.95	81.16 ± 2.87	88.93 ± 1.57	76.73 ± 3.41	96.07 ± 0.87	86.37 ± 5.56
Labeled + 100% Unlabeled	**90.26 ± 1.27**	**86.80 ± 2.07**	**86.57 ± 2.15**	**95.72 ± 0.90**	**81.48 ± 2.81**	**89.27 ± 1.42**	**77.34 ± 3.30**	**96.16 ± 0.86**	**86.71 ± 5.49**

**Table 3 bioengineering-10-00830-t003:** Quantitative comparison with other semi-supervised methods, measured by dice coefficient (mean ± s.d.%). The bolded font represent the experimental results of our model, showcasing its optimal performance in each class.

Method	H	Con	HC	Pne	RO	Line	GGO	Cyst	Avg
MT	88.69 ± 1.63	85.64 ± 2.43	85.07 ± 2.52	94.59 ± 1.08	79.86 ± 3.11	87.72 ± 1.92	75.19 ± 3.62	95.79 ± 0.90	85.17 ± 5.79
UA-MT	88.82 ± 1.60	85.69 ± 2.40	85.26 ± 2.47	94.92 ± 1.07	80.56 ± 3.01	87.54 ± 2.01	75.63 ± 3.57	95.71 ± 0.91	85.48 ± 5.72
CPS-Seg	89.04 ± 1.43	85.78 ± 2.29	85.33 ± 2.43	95.14 ± 1.02	80.52 ± 2.97	88.21 ± 1.74	75.87 ± 3.54	95.60 ± 0.92	85.61 ± 5.65
ST	88.15 ± 1.71	85.10 ± 2.65	84.83 ± 2.61	94.56 ± 1.14	79.81 ± 3.13	87.05 ± 2.09	75.02 ± 3.66	95.54 ± 0.93	84.85 ± 6.07
ST++	89.47 ± 1.42	85.74 ± 2.34	85.71 ± 2.38	95.25 ± 0.98	80.67 ± 2.94	88.69 ± 1.63	76.09 ± 3.50	95.82 ± 0.89	85.89 ± 5.61
ESSegILD (Ours)	**90.26 ± 1.27**	**86.80 ± 2.07**	**86.57 ± 2.15**	**95.72 ± 0.90**	**81.48 ± 2.81**	**89.27 ± 1.42**	**77.34 ± 3.30**	**96.16 ± 0.86**	**86.71 ± 5.49**

## Data Availability

The individual interstitial lung disease (ILD) dataset in this study is available upon request from the corresponding author due to privacy or ethical restrictions.

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
