# Peer review of "Semi-Supervised Segmentation of Interstitial Lung Disease Patterns from CT Images via Self-Training with Selective Re-Training"

_bioengineering, 2023, doi:10.3390/bioengineering10070830_

Round 1

Reviewer 1 Report

I read with interest the paper called "Semi-supervised Segmentation of Interstitial Lung Disease Patterns from CT Images via Self-training with Selective Re-training".

The manuscript contains important information on automatized techniques to aid radiologist in ILD diagnosis. I think it is interesting and well conducted. I only have few comments:

Page 2: don't start a sentence with "But..."; "And...". Introduction needs English revision.

Page 2: "in our work we collected..." this is not appropriate in the introduction section since it contains results. Move to Discussion.

page 12, discussion: the discussion may be expanded with some references to the clinical importance and advantages of implementing such strategy in the medical field.

page 12: conclusion should focus on the final conclusion of your paper based on your hypothesis. please do not to summarize all the results

Introduction needs English revision, 

Author Response

Thank you for your positive feedback on the manuscript. We appreciate your time and effort in reviewing our work. We have carefully considered your comments and suggestions and have made revisions accordingly.

  1. We have polished the introduction section, highlighting the modified parts in red.
  2. The sentence "in our work we collected..." has been moved to the Discussion section as your suggestion.
  3. In the revised manuscript, we have expanded the discussion section by including some references to highlight the clinical importance and advantages of implementing the proposed strategy in the medical field. We believe that these additions will provide a broader context and enhance the significance of our work.
  4. In the revised manuscript, the conclusion focuses on the final conclusion of our paper based on the hypothesis. We provide a concise and clear conclusion that directly addresses our research question and hypothesis.

Reviewer 2 Report

 Major points

In the segmentation model of this study, dropout was used just before the output of models. I don't think that this use of dropout is common in segmentation models.

Label of GGO is wrong in GT of Figure 4. The radiologist's labeling results are not reliable; the GT itself is not reliable. As a result, I am not sure if the entire result of this paper is correct.

What is the criterion for differentiating between bulla and cyst?

How do you distinguish between Reticular opacity and GGO in the labeling?

Minor points

Please cite more papers where ILD is evaluated with AI. Please see the following paper. https://www.ncbi.nlm.nih.gov/pmc/articles/PMC7554005/

Explanation of D_u and D_l should be included in Figure 2

The number of patients should always be added when stating the 11000 CT.

It is not clear whether or not the dataset is divided on the patient basis.

I think P_i=f(u_i), but this should be clearly stated.

If Dice is used as the evaluation metric in Equation 9, then Dice should be used instead of IOU in Equation 8. Why is IOU used in Equation 8?

Table 1 and Table 2 are in the incorrect position.

Author Response

Thank you for your time and effort on the manuscript. We have carefully considered your comments and suggestions and have made revisions in the attached Word file.

Reviewer 3 Report

The figure captions state only the obvious. Making the figure captions more complete and comprehensive would be very useful, as many times readers will be interested just in them and the captions, without necessarily giving full attention to the main text. What one is really curious about when looking at a figure is what is the main message; the most interesting feature that one should look out for, not just what is presented.

Transfer learning is a popular way to generate image representations, and has been used in medical image analysis. Please discuss the methods in 'Deep learning in food category recognition, Information Fusion, 2023, 98: 101859' and 'Detection of abnormal brain in MRI via improved AlexNet and ELM optimized by chaotic bat algorithm'.

The authors need to analyze the potential reasons why the proposed method achieved better results than state-of-the-art approaches.

Please share your codes to the community such as GitHub so that people can follow your research. 

Please provide some drawbacks of your method and some future research directions in Conclusion.

Author Response

(The authors gave the same response as above.)

Round 2

Reviewer 2 Report

My concerns for the labeling is not resolved.

However, it is impossible for me to confirm the accuracy of the labeling.

no comments. 

Reviewer 3 Report

I agree to accept this version.